



# Dual-tracer constraints on the Inverse-Gaussian Transit-time distribution improve the estimation of watermass ages and their temporal trends in the tropical thermocline

Haichao Guo[1], Wolfgang Koeve[1], Andreas Oschlies[1,2], Yan-Chun He[3], Tronje Peer Kemena[1], Lennart Gerke[1], and Iris Kriest[1]

[1]GEOMAR Helmholtz Centre for Ocean Research Kiel, 24148, Kiel, Germany
[2]Kiel University, Kiel, Germany
[3]Nansen Environmental and Remote Sensing Center, Bjerknes Centre for Climate Research, Bergen, Norway

**Correspondence:** Haichao Guo (hguo@geomar.de)

**Abstract.**

Quantifying the mean state and temporal change of seawater age is crucial for understanding the role of ocean circulation and its change in the climate system. One commonly used technique to estimate the water age is the Inverse Gaussian Transit Time Distribution method (IG-TTD), which applies measurements of transient abiotic tracers like chlorofluorocarbon 12 (CFC-

12). Here we use an Earth system model to evaluate how accurately the IG-TTD method infers the mean state and temporal change of true water age from 1981 to 2015 in the tropical thermocline (on isopycnal layer $\sigma_0$=25.5 $\mathrm{kg \cdot m^{-3}}$). To this end, we compared the mean age of IG-TTD ($\Gamma$) derived from simulated CFC-12 with the model "truth", the simulated ideal age. Results show that $\Gamma$ underestimates the ideal age of 46.0 years by up to 50%. We suggest that this discrepancy can be attributed to imperfect assumptions about the shapes of transit-time distribution of water parcels in the tropics and the short atmospheric

history of CFC-12. As for the temporal change of seawater age, when only one transient tracer (CFC-12) is available, $\Gamma$ might be an unreliable indicator and may even be of opposite sign to trends of it due to uncertainties of mixing ratio. The disparity between $\Gamma$ and ideal age temporal trends can be significantly reduced by incorporating an additional abiotic tracer with a different temporal evolution, which we show by constraining $\Gamma$ with sulfur hexafluoride ($SF_6$) in addition to CFC-12.

## 1 Introduction

Ocean ventilation, i.e. the processes transporting surface waters and the associated properties including heat, nutrients, oxygen and other gasses such as carbon dioxide ($CO_2$) to the ocean interior, is of great importance to marine ecosystems and the climate system. Under current transient climate conditions, ventilation slows down the increase in surface air temperature and the rise in atmospheric $CO_2$ by transporting extra heat and $CO_2$ absorbed at the ocean surface into the ocean interior (e.g., Waugh et al., 2004; Sabine et al., 2004; Banks and Gregory, 2006; Sabine and Tanhua, 2010; Khatiwala et al., 2013).

Moreover, circulation supplies oxygen-rich surface water into the aphotic (light-deprived) interior of the ocean, where oxygenic photosynthesis does not occur, thus enables aerobic organisms to thrive and reproduce in the deep ocean. This is particularly





important as many of these organisms, including fish, have significant ecological and economic value, particularly as a source of global food supply.

One method to determine the ventilation strength is via estimating the passage time (the time elapsed since a water parcel was last in contact with the atmosphere, i.e., water age). This property cannot be measured directly in the ocean, but can be diagnosed from measurements of transient abiotic tracers (e.g., Haine and Hall, 2002; Waugh et al., 2004; Fine, 2011; Khatiwala et al., 2012; Sonnerup et al., 2013, 2015; Stöven et al., 2015; Fine et al., 2017; Sonnerup et al., 2019; Jeansson et al., 2023), such as sulphur hexafluoride ($SF_6$) and chlorofluorocarbons (CFCs, e.g., CFC-11, CFC-12) which are purely man-made with well-defined source functions in the atmosphere (Bullister, 2015). The concept of water-mass age has been widely used to estimate the storage of anthropogenic $CO_2$ in the ocean interior (Waugh et al., 2004; Khatiwala et al., 2009), and also in quantifying rates of biogeochemical processes, such as oxygen utilization rates (OUR, Jenkins, 1987; Sonnerup et al., 2013, 2015; Koeve and Kähler, 2016; Guo et al., 2023). Various techniques have been developed to derive water age from these tracers, with the transit time distribution (TTD) method being one of the well-established approaches (Haine and Hall, 2002; Waugh et al., 2003). In contrast to alternative methods like the CFC-12 tracer concentration age (Fine, 2011), the TTD method accounts for the fact that water parcels in the deep ocean comprise fluid elements with different origins and pathways, and thus represents a spectrum of transit times from the ocean surface to the interior location of consideration. One common assumption for the spectrum of individual transit times is the Inverse Gaussian, referred to as IG-TTD (see details in Section 3).

Previous studies using the IG-TTD approach have shown evidence for regional changes in ocean ventilation due to climate change (Waugh et al., 2013; Jeansson et al., 2023). In the Southern Ocean, the water age of the Subantarctic Mode Water (SAMW) has decreased and the age of Circumpolar Deep Water (CDW) has increased over the past decades due to the strengthening and poleward shift of the westerly winds (Waugh et al., 2013). In the Nordic Seas, Jeansson et al. (2023) found overall enhanced ventilation in the upper 1500m from the 1990s to 2010s and reduced ventilation in the deeper waters. Monitoring and understanding these changes is crucial for assessing the consequences on ecosystems and for predicting future responses to climate change.

However, the IG-TTD studies mentioned above usually assume (i) a prescribed constant ratio of mixing and advective transport in the ocean ($\Delta/\Gamma$, i.e. the ratio of width, $\Delta$, and mean, $\Gamma$, of the age spectrum), and (ii) 100% saturation of transient tracers during water mass formation. Uncertainties regarding the validity of these assumptions, can induce uncertainties in the TTD-derived estimates of ventilation and its temporal changes under a changing climate. Even though the limitations of the IG-TTD technique related to the above assumptions have been studied intensively over the past decade (for example, Peacock et al., 2005; Shao et al., 2013, 2016; He et al., 2018; Raimondi et al., 2021, 2023), most of the studies focus only on the high-latitudes and very few focus on the tropical (see details in Section 3). Moreover, available observations of CFC-12 and $SF_6$ in the tropics (Lauvset et al., 2022), can potentially contribute to our understanding on the ocean's ventilation state and its temporal change there. Therefore, we evaluate the IG-TTD technique in the tropical ocean in this study.

While the true (ideal) age in the real ocean can not be measured directly and is hence not known, numerical ocean models provide the possibility to simulate this property as an explicit tracer, together with simulated transient tracers such as CFC-





12 and $SF_6$. Hence, comparing the IG-TTD ages derived from the latter with simulated ideal age provides the possibility to investigate the implication of the assumptions inherent in the IG-TTD method for age estimates. Here we employ an Earth system model (ESM) to investigate whether and to what extent the mean age computed from the IG-TTD method is able to
represent ideal age and its temporal change under global warming in the upper tropical thermocline. Section 2 describes the model, the Flexible Ocean and Climate Infrastructure (FOCI), and the experimental setup. In section 3, the IG-TTD concept and the commonly used assumptions are introduced. Section 4 shows IG-TTD biases and relates these to the applied assumptions. We discuss the IG-TTD application's possible limitations and recommendations for possible improvements in Section 5.

## 2 Model Description

The Flexible Ocean and Climate Infrastructure (FOCI, Matthes et al., 2020) ESM is used in this study. FOCI uses the latest release of the European Centre Hamburg general circulation model (ECHAM6.3) with a nominal resolution of $1.8 \times 1.8$ degree as the atmospheric component. The Nucleus for European Modelling of the Ocean (NEMO3.6, Madec and the NEMO System Team, 2016) is used for its ocean circulation component, coupled with the ocean biogeochemistry model, Model of Oceanic Pelagic Stoichiometry (Kriest and Oschlies, 2015; Chien et al., 2022, MOPS;). The Louvain-la-Neuve sea Ice Model
(LIM2) is the sea-ice module, and the Jena Scheme for Biosphere Atmosphere Coupling in Hamburg (JSBACH) is the land surface component. The ocean and atmosphere components are coupled via the OASIS3-MCT coupler (Valcke, 2013).

  CFC-12 and $SF_6$ are simulated as passive tracers in the NEMO3.6 with a $0.5 \times 0.5$ degree global tripolar grid. The model is vertically discretized on 46 geopotential levels with thickness ranging from 6 m at the surface to 250 m in the deep ocean. A two-step flux-corrected transport, total variance dissipation scheme (Zalesak, 1979, TVD;), is used for tracer advection to
ensure positive-definite values. Tracer diffusion is aligned along isopycnals, with diffusivity set as $600 \ \mathrm{m^2 s^{-1}}$. In the upper ocean, the mixed layer depth is diagnosed by turbulent kinetic energy (TKE) (Blanke and Delecluse, 1993) and vertical mixing is increased for unstable water columns, representing convection in regions of deep and bottom water formation. The relatively coarse resolution does not allow resolving meso-scale eddies, so that the eddy-induced transport is parameterized using the Gent and Mcwilliams (1990) scheme.

An additional tracer, the idealized age tracer, is also included in the ocean component. The idealized age tracer works like a "clock," set to zero at the sea surface and elapsing with one day per day below the surface layer (Thiele and Sarmiento, 1990; England, 1995; Koeve et al., 2015). In other words, the ideal age is the mean age of water parcels that contribute to a grid point in the model but without the constraint of a particular shape (i.e., IG) of the TTD. In this paper, the ideal age serves as a "model truth" of the seawater age.

## 2.1 Experimental set-up

Details of the model set-up have been described by Chien et al. (2022), and we provide only a brief description here. Firstly, the "physics-only" model FOCI was spun up for 1500 years under constant pre-industrial (PI) forcings (e.g., 280 ppm $CO_2$ and fixed aerosol, etc Matthes et al., 2020). The physics-only model was initialized from rest with temperature and salinity



fields from the Polar science center Hydrographic Climatology version 2.1 (PHC2.1; Steele et al., 2001), and sea ice from an
uncoupled ocean-only hindcast simulation (experiment "WEAK 05"; Behrens et al., 2013). After the 1500 years of physical
spin-up, the coupled Earth system model including ocean biogeochemistry was further integrated for 500 years under PI
forcing, followed by a 250 years (drift) period with zero $CO_2$ emissions in which atmospheric carbon dioxide concentrations
were computed prognostically. The biogeochemical tracers phosphate, nitrate, and oxygen were initialized using the World
Ocean Atlas 2013 (WOA2013, Garcia et al., 2013a, b) dataset, and pre-industrial dissolved inorganic carbon and alkalinity
were initialized by GLODAPv2.2016b (Lauvset et al., 2016). The idealized age tracer followed the same spin-up strategy as
the biogeochemical tracers, but was initialized by zeros everywhere in the ocean. Because an age tracer starting from age zero
requires up to several thousand years to reach the equilibrium-state in the ocean (Wunsch and Heimbach, 2008), after the spin-
up of 750 years for fully coupled FOCI, the idealized age tracer is still drifting significantly, but mainly in the deep ocean. At
our focus regions in the upper thermocline, the drift of the idealized age tracer is negligible, as detailed in session 4.1.

Branching off from the spin-up state, we performed the historical and pre-industrial experiments following the Coupled
Model Intercomparison Project 6 (CMIP6) protocol for the emission-driven experiments (Eyring et al., 2016). The historical
(equivalent to *esm-hist* in CMIP6) and pre-industrial control (equvilalent to *esm-piControl* in CMIP6) simulations, were car-
ried out for 165 years, respectively (1850 to 2014). To be brief, the *esm-hist* simulation uses historical $CO_2$ emissions and
prescribed historical time-series of non-$CO_2$ greenhouse gas (e.g., methane, nitrous oxide) concentrations and land-use (Mein-
shausen et al., 2017). The *esm-piControl* simulation is integrated with zero-emission of $CO_2$ and prescribed constant non-$CO_2$
greenhouse gas concentration, land-use that are representative of the earth around the year 1850 (Eyring et al., 2016).

     Implementation of the transient tracers follows the CMIP6 protocol (Orr et al., 2017). Boundary conditions of atmospheric
transient tracers are prescribed for the time period 1936 to 2015 with exactly the same time evolution in the *esm-piControl* and
*esm-hist* runs (Fig. S1; Bullister, 2015), assuming slightly differing tropospheric surface mixing ratios of CFC-12 and $SF_6$ for
the northern and southern hemispheres, respectively. Air-sea gas exchange is computed following Equation (1-2):

$$flux_{\text{a−o}} = (1 - f_{\text{ice}}) \cdot k_{\text{w}} \cdot (C_{\text{equilibrium}} - C_{\text{o}}) \tag{1}$$

$$k_w = a \cdot U^2 \cdot (\frac{Sc}{660})^2 \tag{2}$$

     in which positive $flux_{a-o}$ is the flux from the atmosphere to the ocean and $f_{\text{ice}}$ represents the fraction of sea ice in the
respective grid cell. $C_{\text{equilibrium}}$ is the saturation concentration of the CFC-12 and $SF_6$, determined using sea surface temper-
ature, salinity, and the corresponding atmospheric partial pressure of tracers (Warner and Weiss, 1985; Bullister et al., 2002;
Bullister, 2015). $C_{\text{o}}$ is the simulated concentration of transient tracers in seawater at the ocean surface. The gas transfer veloc-
ity ($k_{\text{w}}$) is determined by a quadratic function from 10m wind speed ($U$), and the revised Schmidt number $Sc$ of individual
transient tracers (Wanninkhof, 1992, 2014). The value of the constant $a$ is
$0.251 \cdot (\text{cm} \cdot \text{h}^{-1}) \cdot (\text{m} \cdot \text{s}^{-1})^{-2}$ (Wanninkhof, 2014)





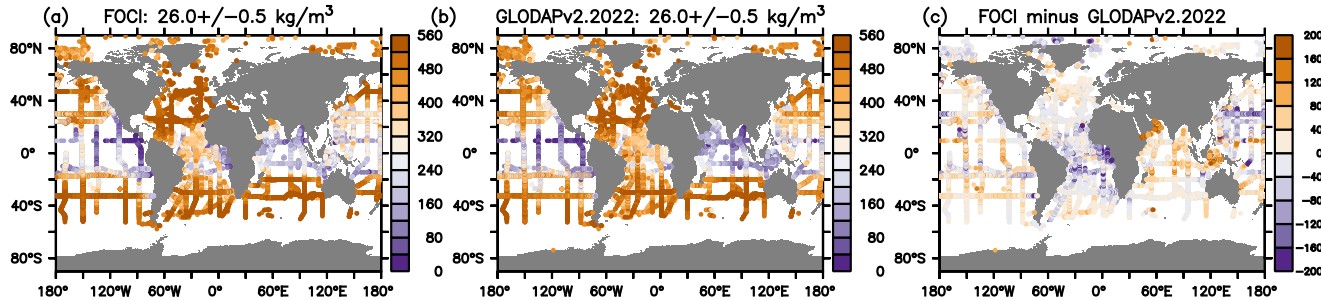

**Figure 1.** Distribution of (a) subsampled simulated CFC-12, (b) observed CFC-12 mixing ratio, and (c) their difference on the isopycnal layer $\sigma_0 = 26.0 \pm 0.5 \, \mathrm{kg \cdot m^{-3}}$, with the unit of parts per trillion (ppt).

## 2.2 Model validation

The skill of FOCI in representing the real ocean ventilation patterns in the thermocline and intermediate waters ($\sigma_0 = 26.0 \pm$

$0.5 \, \mathrm{kg \cdot m^{-3}}$) is evaluated by comparing the mixing ratio of CFC-12 (in units of part per trillion; ppt) in the model ocean and the real ocean. We derived the mixing ratio by dividing the CFC-12 concentration by temperature- and salinity-dependent solubility (Warner and Weiss, 1985). For the real ocean, we used the Global Ocean Data Analysis Project products (GLO-DAPv2.2022: https://www.ncei.noaa.gov/access/ocean-carbon-acidification-data-system/oceans/GLODAPv2_2022/, Lauvset et al., 2022) which provides temperature, salinity, concentrations of transient tracers and also other biogeochemical tracers. For

model simulations, we convert the unit of model output of CFC-12 concentration from $\mathrm{pmol \cdot m^{-3}}$ to $\mathrm{pmol \cdot kg^{-1}}$ (unit in the observational product) by dividing by 1025 $\mathrm{kg \cdot m^{-3}}$, i.e. using a constant density of seawater. Moreover, we subsampled the monthly mean model outputs according to when and where the data were measured.

The broad agreement on CFC-12 mixing ratio distribution between the model simulation and observations in the thermocline and intermediate waters suggests that the model represents the ventilation of the upper ocean reasonably well (Fig. 1). The

pattern correlation coefficient between our subsampled model outputs and observation is 0.917, and the bias is -3.34 ppt (model minus observation, equivent of -0.82% of observation). Both simulations and observations show high partial pressure of CFC-12 in the subtropics (and further north in the Atlantic Ocean) and low partial pressure in the tropics, especially the shadow zones where the water cannot be directly ventilated via outcrops. Relatively high negative biases occur in the tropical South Atlantic and northwest tropical Pacific, which suggest models' underestimation of ventilation in these regions.





## 3 IG-TTD approach

TTD is a distribution of the transit times of water masses from the ocean surface to the interior considering a combination of advective and diffusive transport pathways (Haine and Hall, 2002; Waugh et al., 2003). In brief, the concentration of any passive tracer, $c(\mathbf{r},t)$, at location $\mathbf{r}$ and time $t$ can be related to its surface history and its transit-time distribution, such as:

$$c(\mathbf{r},t) = \int_0^\infty c_0(t-\xi)G(\mathbf{r},\xi)d\xi \tag{3}$$

where $c_0(t-\xi)$ is the history of tracer concentration at the surface ocean $\xi$ years before the time of observation $t$. For CFC-12 and SF6 used in this study, their atmospheric history has been observed from 1936 to 2015 (Bullister, 2015), which provides the boundary condition $c_0$. $G(\mathbf{r},\xi)$ is the transit time distribution at location $\mathbf{r}$. Assuming that the TTD takes the shape of an Inverse Gaussian function (IG) (Waugh et al., 2003), $G(\mathbf{r},\xi)$ can be written as:

$$G(\mathbf{r},\xi) = \sqrt{\frac{\Gamma^3}{4\pi\Delta^2\xi^3}}\exp(\frac{-\Gamma(\xi-\Gamma)^2}{4\Delta\xi}) \tag{4}$$

The shape of $G(\mathbf{r},\xi)$ is determined by the mean age ($\Gamma$), and the variance of the distribution of age scales, i.e. "age spectral width" ($\Delta$), which are expressed as:

$$\Gamma = \int_0^\infty \xi G(\mathbf{r},\xi)d\xi \tag{5}$$

$$\Delta = \frac{1}{2}\int_0^\infty (\xi-\Gamma)^2 G(\mathbf{r},\xi)d\xi \tag{6}$$

Assuming the Inverse Gaussian form (Equation 4) with the two unknown parameters $\Gamma$ and $\Delta$, will imply an unlimited number of ($\Gamma$, $\Delta$) pairs, given a measured interior concentration of a single transient tracer. Making assumptions on the ratio of $\Gamma$ and $\Delta$ will reduce the number of unknowns to 1 and thereby allows estimating both $\Gamma$ and $\Delta$ from a single measurement of transient tracer. $\Delta/\Gamma$ reflects the relative importance of diffusive and advective processes for the ventilated water mass (Waugh et al., 2003). A $\Delta/\Gamma$ of 0 represents a purely advective flow, and $\Delta/\Gamma > 1$ indicates a predominantly diffusive transport. $\Delta/\Gamma$, theoretically, can be constrained by applying two tracers with different input functions coming from their atmospheric time histories (e.g., Waugh et al., 2003; Sonnerup et al., 2015). A pair of tracers measured and used for this purpose is CFC-12 and $SF_6$ (e.g., Tanhua et al., 2008; Stöven et al., 2016). Available observational data of CFC-12 and $SF_6$ suggest that the $\Delta/\Gamma$ ratio ranges from 0-2.0 in the majority of the ocean (e.g., Waugh et al., 2004; Sonnerup et al., 2013). However, it is worth noting that when the $\Delta/\Gamma$ exceeds 1.8, the age becomes highly sensitive to the assumption of tracer saturation levels and cannot be well constrained by the CFC-12/ $SF_6$ pair, as suggested by Stöven et al. (2015).

Additional uncertainty of TTD-based mean age arises from the assumption of constant (usually 100%) saturation of transient tracers at the time of water mass formation (Shao et al., 2013; Stöven et al., 2015; He et al., 2018; Raimondi et al., 2021;



Jeansson et al., 2023; Raimondi et al., 2023). A 100%-saturation assumption is not generally realistic in deep water formation regions, and causes the derived mean age to be biased towards older water ages (He et al., 2018; Raimondi et al., 2021). Also,

the saturation levels of transient tracers in high-latitude regions increased from 1940 onwards (He et al., 2018; Raimondi et al., 2021). These changes include variations in mixed layer depth, growth rates of atmospheric partial pressure, and warming-driven adjustment in solubility (Shao et al., 2013). Considering this rise in saturation levels, water ages range from beeing 0.5 years younger in the upper ocean to 15 years younger in the deep ocean, compared to estimates based on a constant 100% saturation level (He et al., 2018).

Here we calculate the IG-TTD-based mean age from simulated CFC-12, implicating (i, section 4.2) a range of globally homogeneous $\Delta/\Gamma$ values (0.8, 1.0, 1.2, 1.4), (ii, section 4.3) regionally and temporally varying $\Delta/\Gamma$ values constrained by simulated CFC-12 and $SF_6$, and (iii, section 4.4) different saturation level assumptions (100% saturation and time-varying saturation). Instead of choosing one specific point in time, we focus on the time series of $\Gamma$ for the period from 1981 to 2015 when CFC-12 concentrations were intensively measured in the real ocean. We target the isopycnal layer encompassing the

thermocline and intermediate waters ($\sigma_0$=25.5 $kg \cdot m^{-3}$ at about 40-450m depth, Fig. 2c), which is specifically of interest due to the oxygen minimum zones located in this density range. Moreover, the ideal age on this layer is younger than 200 years, which is in the range where CFC-12 based IG-TTD is supposed to be functional with its limited (around 80 years) time history (Sulpis et al., 2021). We use ideal age as reference measure to evaluate whether $\Gamma$ derived from a CFC-12-informed IG-TTD is able to adequately reflect the ventilation processes and their temporal changes.

## 175  4   Results

### 4.1   Idealized age tracer and apparent oxygen utilization

The simulated pre-industrial ventilation pattern via the ideal age on the $\sigma_0$=25.5$kg \cdot m^{-3}$ isopycnal is depicted in Fig. 2a. From the outcrops at mid-latitudes to the tropics, the ideal age increases from 1 yr to 80 yrs. Upwelling, which brings older water from the ocean interior to the upper ocean, might also contribute to elevated ages near the equator. The oldest waters on

the $\sigma_0$=25.5$kg \cdot m^{-3}$ isopycnal with around 150 yrs appear in the eastern North Pacific Ocean where direct contact with the atmosphere is restricted by ocean circulation. This region is known as the ocean "shadow zone" of the ventilated thermocline (Luyten et al., 1983). Apparent oxygen utilization (AOU, the difference between saturated oxygen concentration and measured oxygen concentration) shows a similar spatial pattern as the ideal age, i.e., AOU is high in old waters and vice versa (Fig. 2b). However, low AOU, i.e. oxygen-rich waters are found in the tropical Atlantic Ocean, where the depth of the isopycnal layer

is shallower than 80m and hence in a depth range where oxygen production via photosynthesis may occur. Importantly, the global averages of ideal age and AOU (Fig. 2d, e) show very small drift in *esm-piControl* simulations (-0.016±0.012 $yr \cdot yr^{-1}$ and -0.030±0.031 $mmol \cdot m^{-3} \cdot yr^{-1}$), i.e., the drift of ideal age and AOU is neglectable.



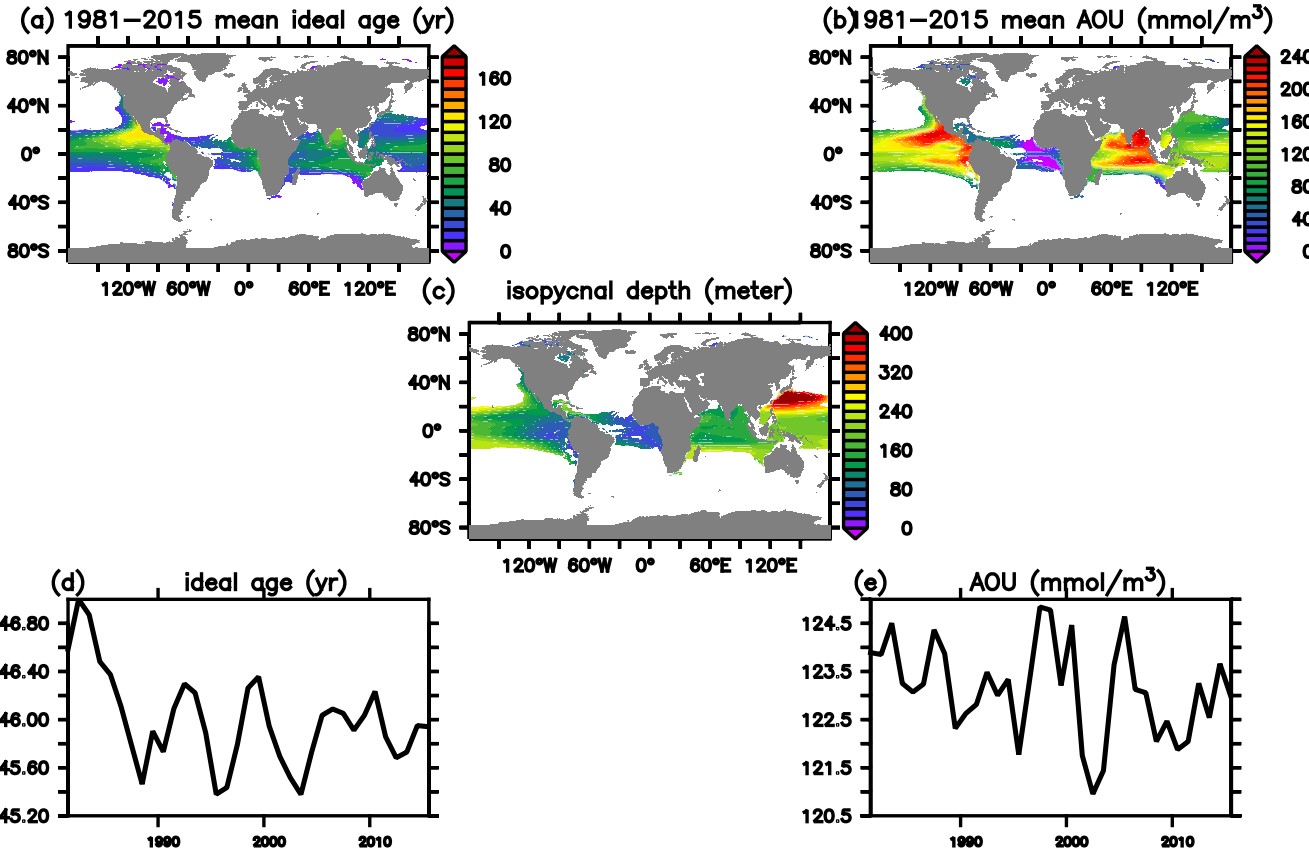

**Figure 2.** Distribution of (a) ideal age (yr), (b) apparent oxygen utilization (AOU, $\mathrm{mmol \cdot m^{-3}}$), and (c) depth (meter) averaged from 1981 to 2015 at isopycnal layer $\sigma_0$=25.5 $\mathrm{kg \cdot m^{-3}}$ in *esm-piControl* simulations. Waters shallower than the local winter mixing depth have been excluded. Panels (d) and (e) present the temporal evolution of ideal age and AOU averaged at the isopycnal layer $\sigma_0$=25.5 $\mathrm{kg \cdot m^{-3}}$ in *esm-piControl* simulation.





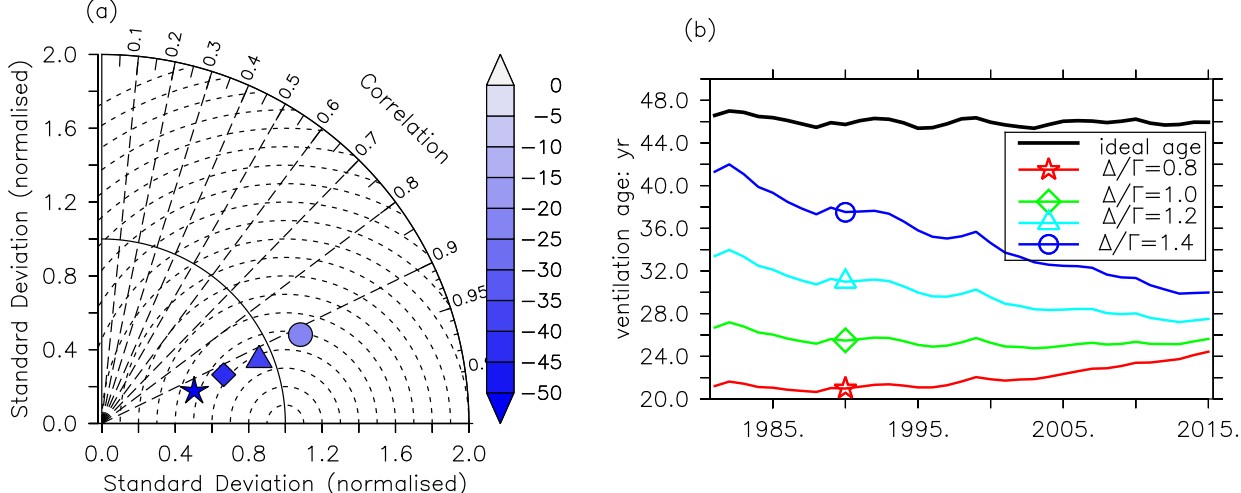

**Figure 3.** In the pre-industrial control run, panel (a) presents the Taylor Diagram between 1981 to 2015 averaged mean age of IG-TTD and the ideal age (as reference) at isopycnal layer $\sigma_0$=25.5 $\mathrm{kg \cdot m^{-3}}$ and the color pattern provides the bias in %. The symbols of star, diamond, triangle and circle indicate $\Delta/\Gamma$ as 0.8, 1.0, 1.2 and 1.4 respectively are applied in IG-TTD calculation. Panel (b) shows the global-averaged ideal age (black), and mean age with $\Delta/\Gamma$ of 0.8 (red), 1.0 (green), 1.2 (cyan), and 1.4 (blue) from 1981 to 2015.

## 4.2 IG-TTD with spatial homogenous $\Delta/\Gamma$ assumptions in *esm-piControl* simulations

Due to the lack of $SF_6$ measurements before 1996 (Lauvset et al., 2022) as well as deliberate tracer release experiments that injected $SF_6$ into the deep ocean and subsequent spreading of this $SF_6$ in parts of the ocean (e.g., the Nordic Sea, Watson et al., 1999; Tanhua et al., 2005; Jeansson et al., 2009), for certain regions only CFC-12 is available for IG-TTD calculation, which implies that fixed spatially homogeneous $\Delta/\Gamma$ ratios are often assumed when computing the IG-TTD, (commonly $\Delta/\Gamma$ = 1±0.2, e.g., Waugh et al., 2004; Jeansson et al., 2023). Here we evaluate the uncertainties of IG-TTD arising from different assumptions of values of $\Delta/\Gamma$ (0.8, 1.0, 1.2, 1.4).

In our *esm-piControl* simulation, the global mean $\Gamma$ inferred from CFC-12 is consistently smaller than the ideal age on the $\sigma_0$=25.5 $\mathrm{kg \cdot m^{-3}}$ isopycnal, regardless of the $\Delta/\Gamma$ assumption (Fig. 3b, Tab. S1). However, the difference between $\Gamma$ and ideal ages varies with different $\Delta/\Gamma$ values. Assuming a low $\Delta/\Gamma$ value (0.8), the horizontally averaged $\Gamma$ value (22.0±2.0 years) is only 48% of the respective ideal age value (46.0±0.8 years). This suggests that the ventilation strength in the upper thermocline is overestimated by the IG-TTD approach for this assumption. On the other hand, assuming a higher $\Delta/\Gamma$ value (1.4), i.e. a more diffusion-dominated transport, the mean $\Gamma$ becomes larger (35.1±6.9 yr), but it is still significantly smaller than the ideal age.

The pattern of 1981-2015 time-averaged $\Gamma$ generally agrees with the distribution of ideal age on $\sigma_0$=25.5 $\mathrm{kg \cdot m^{-3}}$, i.e., low age at subtropical regions and high age at tropical areas (Fig. 4). For all $\Delta/\Gamma$ assumptions, the correlation coefficient between





**Figure 4.** Panels a-d show 1981-2015 averaged mean age of IG-TTD ($\Gamma$) in the pre-industrial control simulation at $\sigma_0$=25.5 $\mathrm{kg \cdot m^{-3}}$, with $\Delta/\Gamma$ of 0.8, 1.0, 1.2, and 1.4, respectively. The $\Gamma$ calculated here is derived from the CFC-12 in *esm-piControl* simulation with a 100% saturation state assumption.



$\Gamma$ and the ideal age is above 0.9 globally. However, the spatial variability of $\Gamma$ substantially differs from the simulated ideal age
and is sensitive to the assumed $\Delta/\Gamma$ (Fig. 3a). More precisely, the spatial variance of $\Gamma$ increases with a higher $\Delta/\Gamma$.

CFC-12 derived $\Gamma$ shows temporal trends from 1981 to 2015 at $\sigma_0$=25.5 $\mathrm{kg \cdot m^{-3}}$ in our *esm-piControl* simulations. Such
trends also depend on the assumed $\Delta/\Gamma$ (Fig. 3b, Tab. S1). Horizontally averaged $\Gamma$ increases at the rate of 0.091±0.016
$\mathrm{yr \cdot yr^{-1}}$ assuming a $\Delta/\Gamma$ of 0.8. Moreover, $\Gamma$ tends to increase more slowly or decrease faster assuming higher $\Delta/\Gamma$. For a
$\Delta/\Gamma$ of 1.4, the global-averaged $\Gamma$ decreases at a rate of -0.340±0.018 $\mathrm{yr \cdot yr^{-1}}$ from 1981 to 2015. However, such temporal
trends in $\Gamma$ are not caused by changing ventilation, because the seawater age is supposed to remain stable in the *esm-piControl*
simulation with stable external forcings. Such constant ventilation is consistent with the much more stable ideal age and AOU
(Fig. 2d,e). To this end, the trends of $\Gamma$ derived from CFC-12 measurements and fixed $\Delta/\Gamma$ should be treated with more caution.

## 4.3 IG-TTD inferred by CFC-12 and SF$_6$ in *esm-piControl* simulations

The tracer pair CFC-12 and SF6 has been used to constrain $\Delta/\Gamma$ in the real ocean when measurements of both tracers were
available and in regions where the impact of SF$_6$ released in local tracer release experiments is negligible (Watson et al., 1999;
Tanhua et al., 2008; Sonnerup et al., 2015; Stöven et al., 2016). To constrain the $\Delta/\Gamma$, we calculate $\Gamma$ separately from simulated
CFC-12 and from SF$_6$ using $\Delta/\Gamma$ from 0.2 to 1.8 for every 0.1 interval, and select the ratio which minimizes the difference
between CFC-12 inferred $\Gamma$ and SF$_6$ inferred $\Gamma$. During the constraining process, we chose the CFC-12 and SF$_6$ simulated in
the *esm-piControl* experiment in the years 2000, 2005, 2010, and 2015, aiming to test the temporal consistency of constrained
$\Delta/\Gamma$. We assume the 100% surface saturation of both tracers for this calculation.

$\Delta/\Gamma$ constrained by both CFC-12 and SF$_6$ in the *esm-piControl* simulations shows substantial spatial differences but some-
how temporal consistency (Fig. 5). Small $\Delta/\Gamma$ ratios, i.e. corresponding to weak water mass mixing, are diagnosed mainly
in the subtropical gyres of the southern hemisphere, and high $\Delta/\Gamma$ ratios are diagnosed in the tropical areas where different
water masses mix on the isopycnal. The temporal consistency of constrained $\Delta/\Gamma$ reflects validity of the essential steady-state
assumption of the TTD method under the pre-industrial condition. It allows for the application of $\Delta/\Gamma$ constrained by CFC-12
and SF$_6$ pair over the past in IG-TTD calculation. For instance, given that SF$_6$ measurements are primarily available post-2000,
while CFC-12 measurements have been extensively recorded since 1980, we can use the CFC-12 and SF$_6$ pair to constrain the
$\Delta/\Gamma$ ratio in 2000 or later. Subsequently, we can employ this ratio, along with CFC-12 measurements before 2000, to calculate
the IG-TTD. However, under transient climate state with internal variablity and underlying trends, the $\Delta/\Gamma$ show more temporal
changes (Fig. S2).

Applying constrained $\Delta/\Gamma$ to the full-time series in the *esm-piControl* simulation still leads to systematic underestimates of
the simulated ideal age by IG-TTD, but helps to find robust temporal trends and variability of the water age (Fig. 6b, Tab. S1).
Results show that global-averaged $\Gamma$ underestimates the simulated ideal age by around 43%, which is similar to the performance
of IG-TTD when using $\Delta/\Gamma$=1. In terms of temporal variability, the correlation coefficient of global-averaged $\Gamma$ and ideal age
is above 0.9. Moreover, the temporal trend of $\Gamma$ overlaps with the trend of ideal age at 95% confidence interval (Tab. S1).



**Figure 5.** $\Delta/\Gamma$ constrained by the simulated concentration of CFC-12 and $SF_6$ on isopycnal layer $\sigma_0$=25.5 $kg \cdot m^{-3}$ under pre-industrial conditions. Panels (a,b,d,e) show $\Delta/\Gamma$ constrained in 2000, 2005, 2010, and 2015, minus the temporal mean $\Delta/\Gamma$ (in panel c). During the calculation, we assume 100% surface saturation of both tracers.




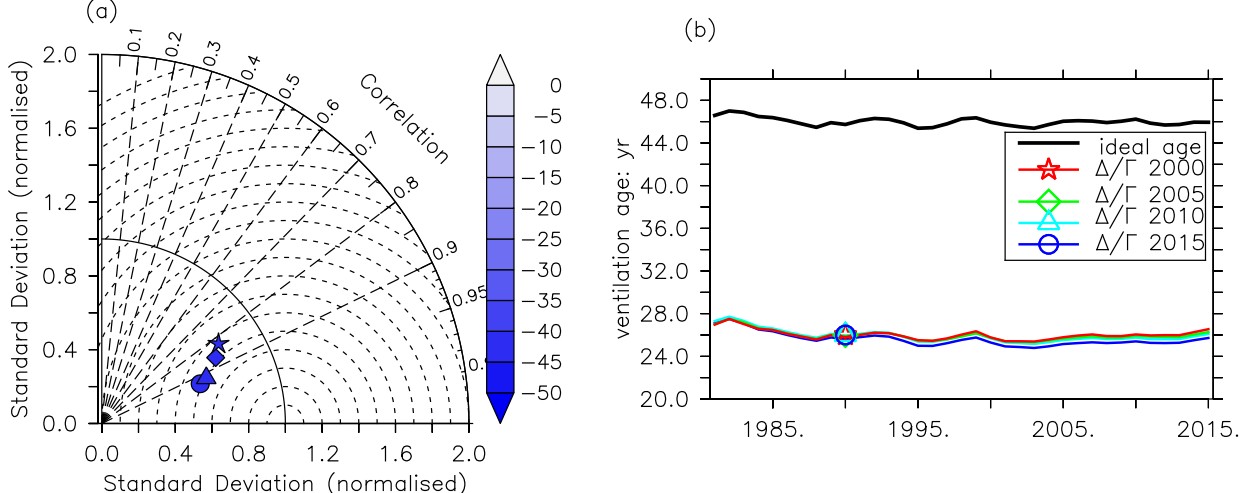

**Figure 6.** The same as Fig. 3, but with different assumption in IG-TTD calculation. Here we use the $\Delta/\Gamma$ constrained by CFC-12 and SF$_6$ in different years. Symbols reveal in which year the $\Delta/\Gamma$ is constrained: star indicates 2000, diamond indicates 2005, triangle indicates 2010, and circle indicates 2015. Panel (b) shows time series of global-averaged ideal age (black), and mean age of IG-TTD with $\Delta/\Gamma$ constrained in 2000 (red), 2005 (green), 2010 (cyan), and 2015 (blue).

## 4.4 IG-TTD with time-varying surface saturation of tracers in *esm-piControl* simulations

As equation (3) suggests, the IG-TTD at location $\mathbf{r}$ is solved with ocean interior tracer concentration $c(\mathbf{r},t)$ and its surface history $c_0(t,\xi)$, where the latter is determined not only from the atmospheric tracer history but also from the tracer saturation state. The latter is usually assumed as 100% (e.g., Waugh et al., 2004); however, this assumption is not everywhere real-
istic, especially in high-latitude regions. Both observations and model simulations have shown considerable undersaturation in deep-convection regions due to the entrainment of older, sub-surface water masses typically with lower concentrations of anthropogenic transient tracers (Tanhua et al., 2008; Shao et al., 2013; He et al., 2018; Raimondi et al., 2021). Moreover, the saturation state also evolves with time due to changes of, e.g., the atmospheric partial pressure of transient tracers, re-entrainment of young waters, oceanic mixed layer depth, and ocean warming. The assumption of tracer saturation constitutes considerable uncertainties in its IG-TTD applications in such regions (Shao et al., 2013; Stöven et al., 2015).

In order to explicitly account for the time-varying saturation of CFC-12 and SF$_6$ in the TTD calculation, we adjusted the mixing ratio histories of CFC-12 and SF$_6$. These adjustments were made based on the simulated saturation of CFC-12 and SF$_6$ at outcrops during hemispheric winter conditions, i.e., March in the northern hemisphere and September in the southern hemisphere, given that water mass formation (ventilation) takes place in early spring. The term "outcrop" here refers to the
place where an isopycnal (or potential density layer) emerges at the surface where exchange with the atmosphere occurs. The CFC-12 and SF$_6$ surface saturation is calculated for $\sigma_0$=25.45 kg$\cdot$m$^{-3}$ to 25.55 kg$\cdot$m$^{-3}$ along the outcrops in the Atlantic Ocean, Pacific Ocean, Indian Ocean, Arctic Ocean, and Southern Ocean. Generally, the saturation state in the Atlantic, Pacific,





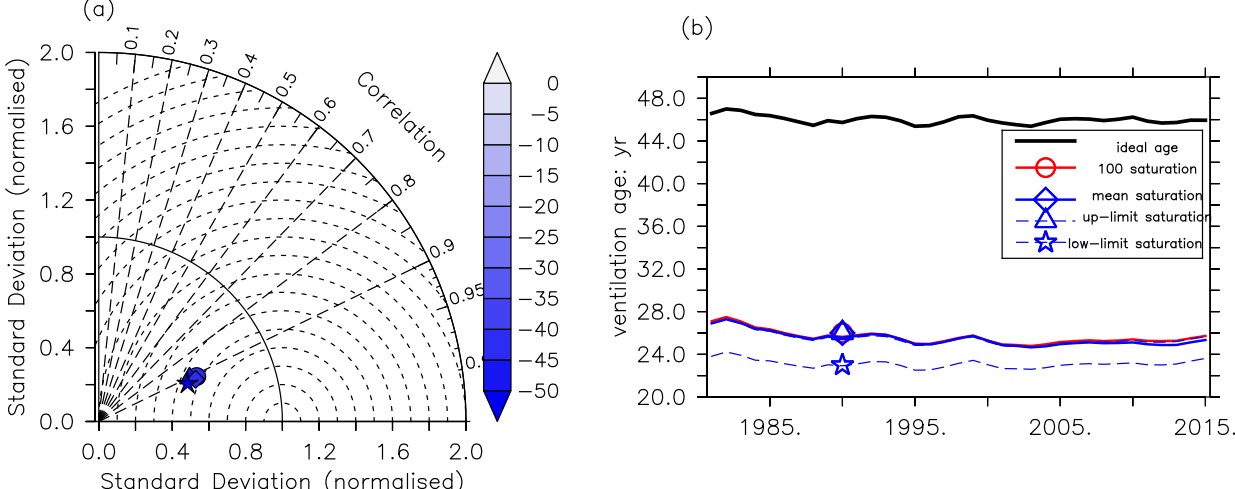

**Figure 7.** The same as Fig. 3, but with different assumption in IG-TTD calculation. Here we use the $\Delta/\Gamma$ constrained by CFC-12 and $SF_6$ in 2015, considering 100% surface saturation (circle) and the time-varying surface saturation of CFC-12 and $SF_6$ (mean saturation state: triangle; up-limit saturation state: diamond; low-limit saturation state: star) in all ocean basins. Panel (b) shows time series of global-averaged ideal age (black), mean age under 100% saturation assumption (red), and the solid values of mean age of IG-TTD considering the mean temporal change of CFC-12 saturation over outcrops (solid blue), change of the upper-limit (upper blue dash) and lower-limit CFC-12 saturation (lower blue dash) saturation state.

and Indian Ocean ranges from 80% to 100% (Fig. S3). Finally, IG-TTD applying adjusted mixing ratio histories of CFC-12 and $SF_6$ are calculated and compared with the ideal age; see details in Text S1.

Considering temporal varying saturation of CFC-12, the $\Gamma$ is systematically younger than the one assuming 100% saturation (Fig. 7), as suggested by other studies (Shao et al., 2013; He et al., 2018). $\Gamma$ calculated with 100% saturation assumption is 0.14 yr (with upper-limit of actual surface saturation) and 2.49 yrs (with lower-limit of actual surface saturation) older than the $\Gamma$ considering temporal varying saturation. The difference between 100%-saturation mean age and temporal varying saturation mean age is supposed to be larger in the deeper ocean because of the more pronounced undersaturation state in the higher

latitude regions (He et al., 2018). Overall the saturation assumption is, however, a minor contributor to the bias of mean age from IG-TTD compared with ideal age in the upper tropical thermocline. This finding aligns with other independent model studies conducted by Shao et al. (2016) and He et al. (2018).

    Moreover, the assumption of 100% surface saturation has minimal impact on the spatial pattern, temporal variability, or the trends of horizontally averaged $\Gamma$ in the upper thermocline $\sigma_0$=25.5 $kg \cdot m^{-3}$ (Fig. 7). The spatial correlation coefficients

between $\Gamma$ and ideal age are all around 0.9 under various assumptions of surface saturation. The Pearson correlation coefficients between the mean age of IG-TTD assuming 100% surface saturation, upper-limit surface saturation, mean surface saturation, lower-limit surface saturation, and ideal age are all above 0.8. Except for $\Gamma$ considering the mean surface saturation of both





tracers, the trends of $\Gamma$ are the same with the temporal trend of ideal age at a 95% confidence interval. Considering the mean surface saturation state of CFC-12 and $SF_6$, $\Gamma$ changes with the rate of -0.050±0.013 $yr \cdot yr^{-1}$, which is slightly different from the trend of ideal age (-0.016±0.012 $yr \cdot yr^{-1}$).

### 4.5 IG-TTD trends in *esm-hist* simulations

One application of the mean age of TTD derived from CFCs and $SF_6$ is to monitor potential changes of ventilation under transient climate conditions, caused for example by changes of wind or upper ocean stratification, as shown in studies by Waugh et al. (2013) for the Southern Ocean and Jeansson et al. (2023) for the Nordic Seas. Here we present the trends of ideal age and TTD-based mean age in our *esm-hist* simulation, where anthropogenic forcings have been incorporated in the fully-coupled FOCI ESM. We assume either spatially homogeneous $\Delta/\Gamma$ ratios (0.8,1.0,1.2,1.4) or CFC-12 and $SF_6$ constrained $\Delta/\Gamma$, together with 100% surface saturation assumption of both tracers to calculate IG-TTD.

Under the assumption of spatial homogeneous $\Delta/\Gamma$ (0.8, 1.0, 1.2, 1.4), our results suggest that $\Gamma$ is not able to detect the ideal age change in *esm-hist* simulation (Fig. 8a, Tab. S2). Ideal age (and AOU) increase with a rate of 0.056±0.019 $yr \cdot yr^{-1}$ (0.122±0.013 $mmol \cdot m^{-3} \cdot yr^{-1}$, not shown) during the period 1981 to 2015, indicating an overall weakening ventilation at the upper isopycnal layer $\sigma_0$=25.5 $kg \cdot m^{-3}$. However, globally averaged $\Gamma$ assuming spatially homogeneous $\Delta/\Gamma$ changes with rates of 0.115±0.020 $yr \cdot yr^{-1}$ ($\Delta/\Gamma$=0.8), -0.009±0.020 $yr \cdot yr^{-1}$ ($\Delta/\Gamma$=1.0), -0.151 ±0.022 $yr \cdot yr^{-1}$ ($\Delta/\Gamma$=1.2), and -0.313±0.026 $yr \cdot yr^{-1}$ ($\Delta/\Gamma$=1.4), i.e, temporal trends of $\Gamma$ significantly differ from that of the ideal age in the *esm-hist* simulation (Tab. S2). Our earlier finding of strong trends of $\Gamma$ under stable climate conditions (*esm-piControl* experiment,Fig. 3b, Tab. S1), where no trend is expected, supports the conclusion that the respective single tracer $\Gamma$ trends in *esm-hist* are artificial and unreliable.

Using $\Delta/\Gamma$ constrained by the CFC-12 and $SF_6$ tracer pair in *esm-hist* simulations, the difference between temporal trends of globally averaged $\Gamma$ and ideal age has significantly narrowed down (Fig. 8b, Tab. S2). Using $\Delta/\Gamma$ constrained in 2000 and 2005, the temporal trends of $\Gamma$ are 0.064±0.019 $yr \cdot yr^{-1}$ and 0.055±0.020 $yr \cdot yr^{-1}$ respectively, which is in line with the temporal trend of ideal age at a 95% confidence interval. Besides, the Person correlation coefficients between respective globally-averaged $\Gamma$ and ideal age are 0.947 and 0.949. In other words, the mean age from constrained IG-TTD is able to detect the temporal variability and trend of the simulated ideal age. While, when using $\Delta/\Gamma$ constrained in 2010 and 2015, global-averaged $\Gamma$ indicate no significant trend (0.017±0.021 $yr \cdot yr^{-1}$ and -0.016±0.021 $yr \cdot yr^{-1}$), but still performs better than when using spatially homogeneous $\Delta/\Gamma$.

## 5 Discussion

Over the last four decades, extensive measurements of dissolved anthropogenic gases, such as CFCs and $SF_6$, have been conducted in the global ocean (Lauvset et al., 2022) through programs like the World Ocean Circulation Experiment (WOCE) program and the Global Ocean Ship-Based Hydrographic Investigation Program (GO-SHIP). These measurements provide valuable data for diagnosing the spatial pattern and temporal changes in seawater age, anthropogenic carbon storage, and





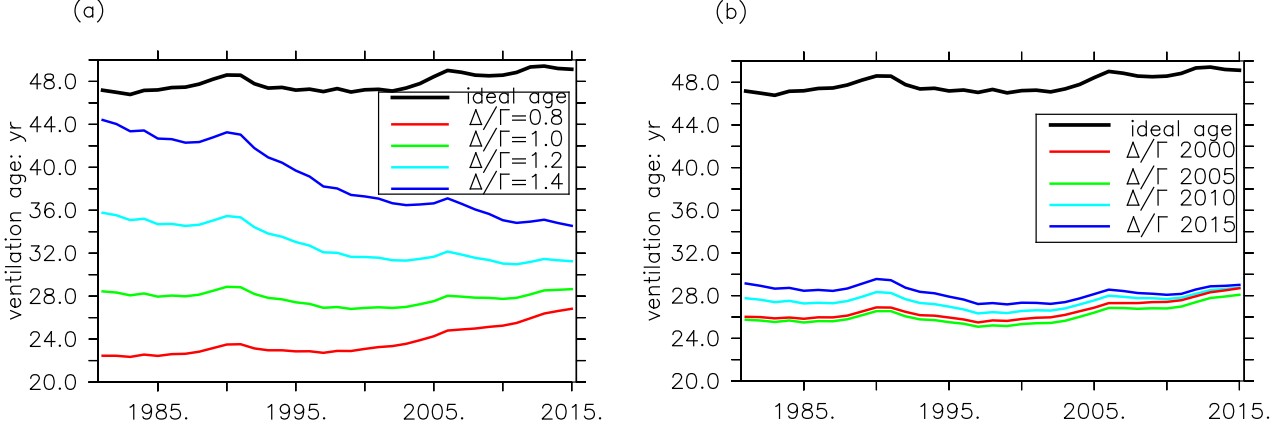

**Figure 8.** Trends of globally-averaged ideal age and $\Gamma$ for various assumptions about $\Delta/\Gamma$ for experiment *esm-hist* at the density layer $\sigma_0$=25.5 $\mathrm{kg \cdot m^{-3}}$. (a) Globally-averaged ideal age (black) and mean age from CFC-12 based IG-TTD using spatial homogenous $\Delta/\Gamma$ (red: 0.8, green: 1.0, cyan: 1.2, and blue: 1.4) from 1981 to 2015 in historical simulation. (b) Global-averaged ideal age (black) and mean age of IG-TTD using CFC-12 and $\mathrm{SF_6}$ constrained $\Delta/\Gamma$. In panel (b), the line color represents the year when the $\Delta/\Gamma$ ratio has been constrained (red: 2000, green: 2005, cyan: 2010, blue: 2015). 100% surface saturation of both CFC-12 and $\mathrm{SF_6}$ is assumed for all calculations.

biogeochemical processes, such as respiration, using the IG-TTD technique. A recent study by Jeansson et al. (2023) shows overall stronger ventilation in the Greenland Sea from the 1990s to the 2010s, in line with the results revealed by AOU. As for biogeochemical processes, many field studies use mean age of IG-TTD as the seawater age (Álvarez-Salgado et al., 2014; Sonnerup et al., 2013, 2015; Sulpis et al., 2023) to diagnose regionally averaged oceanic aerobic respiration rates by calculating OUR, i.e. the slope of the least square regression of AOU and seawater age on potential density surfaces (Jenkins, 1987; Guo
et al., 2023).

Nevertheless, uncertainties associated with the IG-TTD technique, including its ability to accurately represent climatological mean, spatial variance, and temporal trends of "true" seawater age, may hinder its direct applicability. Our study suggests that across all tested assumptions, the mean age derived from the IG-TTD method tends to underestimate the ideal age within the upper thermocline. This discrepancy could potentially affect the estimation of ocean heat uptake and anthropogenic carbon
storage when relying on IG-TTD mean ages in certain oceanic regions. Additionally, uncertainties of the spatial variance of the mean age of IG-TTD might further limit the application for OUR computation, as OUR is calculated from spatial gradients of $\Gamma$ and AOU in the real ocean. The third caveat of IG-TTD technique is that the temporal trend of the mean age might not faithfully reflect the temporal changes of ocean ventilation unless the $\Delta/\Gamma$ can be constrained by dual transient tracers (whenever available). We have shown that applying an additional tracer ($\mathrm{SF_6}$) with a different input function in the IG-TTD
technique improves the estimate of temporal change of ocean ventilation strength. Moreover, in the Nordic Sea where $\mathrm{SF_6}$ has



been deliberately released in the past, potential future measurements of Argon-39 combined with CFC-12 measurements may be used to constrain the $\Delta/\Gamma$ in local (Ebser et al., 2018).

Why the mean age of IG-TTD tends to underestimate the ideal age? Firstly, the IG-TTD technique assumes a spectrum of transit-time scales of water parcels following an unimodal IG distribution. However, this might not be an accurate assumption
in ocean regions with complex circulation. Studies by Haine and Hall (2002), Peacock and Maltrud (2006), and Shao et al. (2016) suggest that TTD distributions often exhibit a bimodal or multimodal structure in upwelling and downwelling regions, as well as in regions where multiple source water masses converge (such as the equatorial region and the Southern Ocean). For example, Peacock and Maltrud (2006) compared the distribution of CFC-like tracer in the ocean based on the model-simulated TTD ("actual" CFC) and on the TTD obtained using the Inverse Gaussian with $\Gamma$ and $\Delta$ of the model simulation ("predicted"),
and found that "predicted" CFC-like concentrations by IG-TTD are only half of "actual" values at a depth of 245 meters in the tropical regions (see their Fig. 13). In other words, the mean age of IG-TTD derived from the directly simulated CFC-like tracer is younger than the simulated real water age. To capture such distributions, linear combinations of IG distributions can be used (Equation 7), as proposed by Waugh et al. (2003). However, incorporating additional source water masses in the analysis requires additional transient tracers with different atmospheric histories to constrain the $\Delta/\Gamma$ ratios, as well as the respective
water fractions characterizing the multimodal IG distribution, as mentioned by Stöven et al. (2016). With a limited number of transient tracers available from observations and in our model experiments, the multimodal TTD cannot be solved in this study.

Another potential reason for the disparity of ideal age and mean age of IG-TTD is the limited atmospheric history length of CFC-12 and $SF_6$ (Shao et al., 2016). These gases only began to be measurable in the atmosphere after 1936 and 1953 respectively, providing a historical span of only 88 years for CFC-12 and 71 years for $SF_6$ until now (Bullister, 2015). Therefore, the
extended tail of older ages in the spectrum cannot be adequately constrained by these tracers alone. Incorporating additional age tracers with longer atmospheric histories or lifetimes, such as Argon-39 with a half-life of 269 years, might offer better constraints on older water components in the transient age spectrum and help reduce the disparity of IG-TTD and ideal age. To provide more details, a set of simulations of CFC-12, $SF_6$, Argon-39, and ideal age is required.

$$G(\mathbf{r}, \xi) = \alpha_1 G_1(\Gamma_1, \Delta_1) + \alpha_2 G_2(\Gamma_2, \Delta_2) + ... + \alpha_n G_n(\Gamma_n, \Delta_n) \tag{7}$$

Continuous measurements of CFC-12, $SF_6$, and hopefully additional transient tracers (e.g., $^{39}Ar$) with different and longer
atmospheric history, together with techniques of separating source water masses, might help to quantify ocean ventilation better. Our study confirms the better performance of IG-TTD when dual transient tracers (CFC-12 and $SF_6$) are implicated compared to the case of using only a single tracer (CFC-12). Together with measurements of transient tracers that have longer atmospheric histories might further improve the performance of TTD in representing old waters (Shao et al., 2016). Water fractions, required in equation 7, can be derived by applying the Optimal Multi-Parameter (OMP) analysis (Karstensen and
Tomczak, 1998).



## 6 Conclusion

Our study evaluates the Inverse Gaussian Transit Time Distribution (IG-TTD) method by comparing the mean age of IG-TTD ($\Gamma$) and a ground truth measure of water age, the ideal age on the isopycnal layer $\sigma_0$=25.5 $\mathrm{kg \cdot m^{-3}}$ within an Earth system model. Our results suggest that:

(i) $\Gamma$ substantially underestimates the ideal age of 46.0 years by up to 24.3 years. Such a difference might arise from the assumption that the transit-time distribution of water parcels follows the unimodal Inverse Gaussian distribution, and also from the cut-off of the long-tail of old ages in the spectrum due to the limited atmospheric history length of CFC-12 and $\mathrm{SF_6}$.

    (ii) Possibly incorrect assumptions about surface saturation of CFC-12 and $\mathrm{SF_6}$ do not resolve the discrepancy between $\Gamma$ and ideal age.

(iii) When only CFC-12 is used (assuming spatial homogeneous $\Delta/\Gamma$ of 0.8, 1.0, 1.2, 1.4), the derived $\Gamma$ shows significantly different trends from 1981 to 2015 compared to the ideal age, indicating the misrepresentation of temporal change of ocean ventilation inferred by IG-TTD technique in this case.

    (iv) When dual tracers (CFC-12 and $\mathrm{SF_6}$) are used in the IG-TTD techniques, $\Gamma$ can correctly detect temporal variability and trend of true water age in the upper tropical thermocline.

*Code and data availability.* The data and material that support the findings of this study are available through GEOMAR at: https://data. geomar.de/downloads/20.500.12085/b5baa5f6-5bda-458f-bfaf-3da3b789a972/ (Guo et al., 2024).

*Author contributions.* IK and WK conceptualized the research. HG carried out the analysis. All authors discussed the results and wrote the manuscript.

*Competing interests.* The contact author has declared that none of the authors has any competing interests.

*Acknowledgements.* We acknowledge the work of Heiner Dietze regarding coding of CFC-12 and $\mathrm{SF_6}$, and constructive discussions with colleagues from the Biogeochemical Modelling research unit at GEOMAR Helmholtz Centre for Ocean Research Kiel. We especially thank Toste Tanhua and Lorenza Raimondi for the code for calculating Inverse Gaussian transit time distributions. We also wish to acknowledge use of the Ferret program of NOAA's Pacific Marine Environmental Laboratory for analysis and graphics featured in this paper.



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
