# Peer review of "Dual-tracer constraints on the Inverse-Gaussian Transit-time distribution improve the estimation of water mass ages and their temporal trends in the tropical thermocline"

_EGUsphere, 2024_

## Author Comment (AC1)

The authors thank the reviewer for your very helpful comments and suggestions. We here provide a point-by-point response to the reviewer. The reviewer's comments are given in black, and our responses are given in blue color. We refer to line numbers in the revised manuscript.
* * *
Reviewer #1 Comments to Author:

This is a fine effort that provides guidance on how realistic changes in ventilation inferred from repeated CFC (and SF6) sections can be, and thus this can be an important contribution. It also focusses on the tropics, rather than the usual subtropical gyres. The work is careful and well focussed. I applaud the authors for their focus on saturation level uncertainties and trends. That said, I was disappointed that there was not much effort devoted to real-world scenarios. Researchers don't go out and measure a mean CFC-12 and a mean SF6 on a global isopycnal, they go out and measure discrete water samples. From those discrete samples, they attempt, perhaps optimistically, to infer basin-wide patterns of changes in ventilation. I suspect an 'upgrade' is easily within reach for these authors - choosing a location or two (or a section or two) and explicitly calculating the change in ventilation inferred from tracers as compared with the actual change. I would not insist on this happening - the work is fine as it is - but it may improve the impact of their paper.

A: We appreciate your thoughtful and constructive feedback on our work. Our current study is centered on evaluating how the inherent assumptions—such as mixing ratio and saturation—within the IG-TTD framework impact its accuracy in reflecting real temporal changes in ocean ventilation. We understand and acknowledge your point that the discreteness of measurements can lead to additional uncertainties of real-world estimates of ocean ventilation change on a basin-wide scale. However, addressing this issue is beyond the scope of our current research. The sampling issue is, actually, part of another piece of our work (currently under review), which implicitly builds on the findings of this manuscript and explores the change in ocean ventilation in the North Atlantic over the past three decades (https://www.researchsquare.com/article/rs-5595029/v1). In that follow-up work, we compared the subsampled model outputs (i.e., model outputs subsampled analogously based on the timing and location of available observations) and complete model outputs in 7 Earth System Models (compare in that manuscript: lines 87-99, and lines 158-170). We found that the spatial and temporal coverage of CFC-12/SF6 measurements in the North Atlantic is adequate for indicating real ocean ventilation changes. Noteworthy is that the North Atlantic is a region where CFC-12/SF6 is sampled relatively well; therefore, for less well sampled regions such as the tropical Pacific, we would need further investigations in the future.

Line 153. Consider rewording this. The lessons from Waugh et al 2003 and from Stoeven et al 2015 are that CFCs and SF$_6$ only provide meaningful constraints on Delta/Gamma when <= 1.8 or so. Those papers do not indicate that the real Delta/Gamma in the ocean is <= 1.8, merely that the tracer pair doesn't provide useful information >= 1.8. Delta/Gamma could be 3, it could be 6, but this tracer combination wouldn't tell us that, and wouldn't be able to distinguish 3 from 6.

A: We are grateful that the reviewer pointed it out. We have clarified this in our revised manuscript according to your suggestions. We deleted the sentence, "Available observational data of CFC-12 and SF6 suggest that the Delta/Gamma ranges from 0-2.0 in the majority of the ocean (Waugh et al., 2003; Stöven et al., 2015)". Instead, we write now "Theoretically, the

Delta/Gamma ratio can vary over a large range in nature. However, when Delta/Gamma exceeds 1.8, the water age cannot be well constrained by the CFC-12/SF6 pair (Waugh et al., 2003; Stöven et al., 2015). Effectively, the search range of Delta/Gamma is therefore restricted to be smaller than 1.8, which is also the case in this study." (lines 153-155 in the revised version)

Figs 2, 4 &5.  It is very hard to see the patters described in the text.  This paper focusses on the tropics.  One way to improve visibility would be to remove latitudes > 40 (maybe even 35?)

A: We thank the reviewer for this suggestion! We have modified the figures as suggested, limiting latitudes to 35S - 35N. We show Fig. 2 as an example here.

[Figure]

Fig. 2: Distribution of (a) ideal age (yr), (b) apparent oxygen utilization (AOU, mmol·m$^{-3}$), and (c) depth (meter) averaged from 1981 to 2015 at isopycnal layer $\sigma_0$=25.5 kg·m$^{-3}$ in *esm-piControl* simulations. Waters shallower than the local winter mixing depth have been excluded. Panel (d) presents the temporal evolution of ideal age (red line, with left y-axis) and AOU (blue line, with right y-axis) averaged at isopycnal layer $\sigma_0$=25.5 kg·m$^{-3}$ in *esm-piControl* simulation.

Relatedly.  Fig. 3b.  Do these results vary by basin?  The processes and pathways bringing tracers to the eastern tropical Pacific are quite different from, say, the Arabian Sea.  This figure shows the mean.

I have the same comment about Figure 6b and 7b.  Okay maybe the twin-tracer TTDs *can* do a good job representing changes in the global mean ventilation of this isopycnal.  But people don't go out and measure a global mean pSF6 and pCFC-12, they have section measurements from which they hope to derive larger-scale conclusions.  How would these three figures look at some individual, spot locations or sections?  Do these conclusions, which hold for the isopycnal mean, apply universally in individual regions?

A: We appreciate the suggestion and have provided an additional figure in the revised manuscript (Fig. 9). This figure shows the temporal trend of ideal age and mean age of IG-TTD under different mixing ratio assumptions at each grid box. We described this figure in the revised manuscript as below (lines 298-304 in the revised manuscript). In short, the dual-tracer constrained IG-TTD performs better in presenting ideal age trends in all ocean basins compared with a single-tracer constrained IG-TTD.

"Noteworthy, the dual-tracer constrained IG-TTD demonstrates superior performance in discerning spatial patterns and magnitude of temporal changes in ideal age compared to the single-tracer constrained IG-TTD (Fig.9). The single-tracer constrained IG-TTD is very sensitive to the chosen value of Delta/Gamma, commonly showing a spurious age increase with low values of Delta/Gamma and an age decrease with high values of Delta/Gamma across all ocean basins (Fig.9c-f). While the dual-tracer method exhibits some spurious trends in the eastern tropical Atlantic and western tropical Indian Ocean, it generally provides a more accurate representation of the spatial patterns and magnitudes of true ideal age trends. Notably, it correctly identifies regions with no significant trends in ideal age (Fig.9a,b)."

[Figure]

Fig.9: Temporal trend of ideal age (a) and mean age of IG-TTD with different assumptions on the value of delta/gamma (b-f) on isopycnal layer $\sigma_0$= 25.5 kg/m3 in the *esm-hist* simulation. Stippling designates areas where the ratio of standard deviation and mean of the regression slope of ideal age (or mean age) against time exceeds 1 (i.e., no significant trends).

Figure 4.  It would be very useful and interesting to see a figure of the percent mismatch of at least one of these (your choice) from the reference, figure 2a.

A: We have provided an additional figure (Fig. S2) showing the percent mismatch between mean age computed with different assumptions about the value of delta/gamma and the simulated ideal age (by using the ratio (mean age)/(ideal age) x 100-100). To be brief, besides the old waters in the northeast tropical Pacific and the Bay of Bengal with Delta/Gamma values higher than 1.0, the mean age of IG-TTD underestimates the ideal age. Moreover, the relative difference between ideal age and mean age is slightly higher in young waters (lines 200-202).

[Figure]

Fig. S2: Panel (a) shows the distribution of ideal age (yr) averaged from 1981 to 2015 on isopycnal layer $\sigma_0$= 25.5 kg/m3 in the esm-piControl simulation. Panels (b-f)  show the percent

mismatch between mean age with different assumptions about the value of delta/gamma and the simulated ideal age (by using (mean age)/(ideal age) × 100-100).

Line 233 – This is very similar to the findings of Mecking et al 2004 based on simple pCFC ages in the early 1990s– and to the findings of Thiele and Sarmiento (1990) based on pCFC ratio ages in the 1980s. Some attribution to these pioneers, somewhere in this paper, would seem appropriate.

A: We agree, and have added those references in our discussion part on the IG-TTD based mean age being younger than the ideal age.

9. uncertainties in the mixing ratio,

A: We have modified the sentence according to the reviewer's suggestion. (done)

26-28. One citation will suffice – I'd recommend Jenkins 1980 and Weiss et al 1985

A: We agree and have added these references as recommended and removed the rest. (done)

30. Gruber 1998

A: We have added the reference as recommended. (Line 29)

34. Doney and Bullister 1992 rather than Fine 2011

A: We have modified the reference as recommended. (Line 33)

36 … individual transit times is that which results from one-dimensional transport and mixing, the Inverse Gaussian…

A: We have modified the sentence as suggested. (Line 36)

51 most of the studies focus on the subtropical gyres, and very few focus on the tropics.

A: We have clarified this in the revised version. (Line 50)

60 … represent ideal age and temporal changes in ideal age…. (cumbersome but clearer)

A: We thank you for the language suggestion and have modified the sentence in the revised version. (Line 57)

147 …from measurement of a single transient tracer

A: We have modified the sentence accordingly. (Line 147)

159 - Tricky – the saturation level mis-assumption causes the derived age to be older than it should, however it is still younger than the ideal age (or, water age)

Maybe try wording like

…causes the tracer-derived age to be older than with a more realistic, undersaturated, boundary condition.

A: We thank you for pointing this out and have reworded as suggested. (lines 158-160)

160 Shao et al 2013 should be cited here as well.

A: We have added the reference as recommended. (lines 160)

162 …rise in saturation levels, TTD-derived water ages range from being…

A:  We modified the sentence as suggested (line 163).

192 …calculation, so fixed spatially homogeneous…

A:  We modified the sentence as suggested (line 191).

205  More precisely, the spatial variance of Gamma increases with higher Delta/Gamma, with 1.2 being most representative of the spatial variation in ideal age.

       (If I'm interpreting the Fig. 3a correctly, ignore this remark if I'm not)

A: The interpretation is correct and we have added this information in the corresponding lines (lines 206-207)

251-252.  …along outcrops and in the Atlantic, Pacific and Indian Ocean ranges from 80% to 100% (Fig. S3).

A: We have modified the sentence as suggested (lines 252-254).

262 This is an important result. I would start the sentence 'Notably, the assumption…'

A: We have emphasized this result by modifying the sentence.

278  An even more important result. Possibly clarify a bit …our results suggest that Gamma constrained by CFC-12 alone is not able…

A: We have clarified this in the revised version.

325  Does this also mean that 'ages' from those concentrations would be about half the ideal age? That result is very similar to yours, and similar to those of Mecking et al 2004.

A: We appreciate the reference of Mecking et al., 2004 and we have added this reference in our revised manuscript. To our understanding,  Peacock and Maltrud (2006) compared the concentration of CFC-like tracer based on the model-simulated TTD ("actual" CFC) and on the TTD obtained using the Inverse Gaussian with the same gamma and delta of the model simulation ("predicted"). They found that "predicted" CFC-like concentrations are around **half** of "actual" values at a depth of 245 meters in tropical regions. In other words,  to achieve the same abiotic transient tracer concentration inferred from model-simulated TTD, the IG-TTD would need to reduce its mean age,  i.e., the mean age of IG-TTD is smaller the mean age of the "real" TTD with the same abiotic tracer concentration. However, **half** the CFC concentration does not correspond to twice the mean age. To answer whether the mean age of IG-TTD underestimates the mean age of TTD by **half**, it requires the following procedure: (i) calculate CFC-tracer

concentration based on simulated TTD; (ii) calculate IG-TTD from calculated CFC-tracer concentration; (iii) compare mean age of TTD and mean age of IG-TTD.

We conducted a straightforward test by calculating the mean age of IG-TTD, assuming CFC-12 partial pressures of 344 ppt and 172 ppt (half) in the Northern Hemisphere in 2009 (shown by the figure below). During the calculations, we considered both low and middle Delta/Gamma ratios. With a Delta/Gamma ratio of 0.2, the mean ages of IG-TTD derived from the high and low partial pressures were 28 years and 40 years, respectively, indicating that the mean age with low partial pressure is over half of that with high partial pressure. Conversely, with a Delta/Gamma ratio of 1.0, the mean ages were 40 years and 100 years, respectively, meaning the mean age with low partial pressure is less than half of that with high partial pressure.

[Figure]

Not shown in manuscript: The thick black line in the left panels represents the input function of CFC-12 partial pressure (pCFC-12, in ppt). The blue and red circles represent 172 ppt and 344 ppt in 2009, respectively. The lines in the left panel represent the expected pCFC-12 evolution when mean age does not change. Right panels illustrate the Inverse Gaussian Transit Time Distribution (IG-TTD), and the dash vertical lines represent the mean age.

333. Rather than Shao et al I'd choose Waugh et al 2003 as the citation

A: We have changed the reference to Waugh et al. (2003).

333. 'only began to be measurable in the atmosphere after 1936 and 1953'

These weren't really measured back then, they were invented and presumably released into the atmosphere around those dates. Bullister's reconstruction is a model using CFC and SF6 production and assuming some release function that is tuned to CFC SF6 measurements when those started to be readily available – 1970s?

Rather than muddy the waters, you could just write 'These gases were released into the atmosphere after 1936 and 1953, respectively…'

A: We have modified the sentence as suggested.

337 – A set of 39-Ar simulations would be nice, but what would be nicer still is a set of 39-Ar *measurements.* For that study, when you do it, you may need to focus your efforts on waters that are a bit older. Good luck!

A: Thanks a lot! At the Ocean Science Meeting 2024, there was an insightful talk about 39-Ar, highlighting advancements in techniques that significantly improve the efficiency of 39-Ar measurement and reduce the necessary water sample size. Although I'm uncertain about when large-scale measurement of 39-Ar will become available, I hope it will be soon. We agree that, due to its longer half-life time of 269 years, 39-Ar can in particular improve the quantification of deep ocean ventilation.

Fig. 8 – I can't help but notice that all of these 'data' treatments capture the brief slowdown events of 1990 and 2006. Whether real-world sampling would capture this is another question, but you might note in the text that these events are reflected in the tracer concentrations (and in their derived ages).

A: We agree that all mean age effectively captures the slowdown of ventilation in the years 1990 and 2006, as evidenced by the noticeable bulge in both the ideal age and mean age metrics. We have noted, "The averaged gamma effectively captures the slowdown of ventilation in the years 1990 and 2006 (the little bulge of ideal age and all mean age) " in our revised manuscript.

Citations provided by reviewer

Doney, S. C., and J. L. Bullister (1992) A chlorofluorocarbon section in the eastern North Atlantic, Deep Sea Research 39, 11-12, 1857-1883.

Gruber, N., 1998 Anthropogenic CO2 in the Atlantic Ocean, Glob. Biogeochem. Cycles 12, 1, doi.org/10.1029/97GB03658

Jenkins, William J.. 1980. "Tritium and 3He in the Sargasso Sea." *Journal of Marine Research* 38, (3). https://elischolar.library.yale.edu/journal_of_marine_research/1518

Mecking, S., M. J. Warner, C. E. Greene, S. L. Hautala, and R. E. Sonnerup 2004 Influence of mixing on CFC uptake and CFC ages in the North Pacific thermocline, J. Geophys. Res. Oceans 109, C7, https://doi.org/10.1029/2003JC001988

Weiss, R., Bullister, J., Gammon, R. *et al.* Atmospheric chlorofluoromethanes in the deep equatorial Atlantic. *Nature* 314, 608–610 (1985). https://doi.org/10.1038/314608a0

---

## Author Comment (AC2)

The authors thank the reviewer for your very helpful comments and suggestions. We here provide a point-by-point response. The reviewer's comments are given in black, and our responses are given in blue color.
* * *
Reviewer #2 Comments to Author:

Review on 'Dual-tracer constraints on the Inverse-Gaussian Transit-time distribution improve the estimation of watermass ages and their temporal trends in the tropical thermocline' by Haichao Guo et al.

The aim of this study is to compare the 'real' mean (or ideal) ages with the mean ages inferred from Inverse Gaussian (IG) functions for the isopycnal sigma_theta=25.5 (including thermocline and intermediate waters) over the period 1981-2015. The ideal age cannot be observed, but the IG functions can be inferred form the observations of anthropogenic tracers like CFCs and SF6. Hence, it is of interest, in how far these observational inferred mean ages agree with the 'theoretical' ideal age. This can only be tested in a model study. The authors use the FOCI model to simulate mean age, CFC-12 and SF6. After a short model evaluation, the IG functions are inferred for different cases: from the modeled CFC-12 data alone, assuming fixed Delta/Gamma ratios, and by inferring both IG parameters Delta and Gamma from the modeled SF6 and CFC-12 fields. The IG parameter Gamma (mean age) and its temporal change between 1981 and 2015 is compared with the modeled ideal age.

This comparison of the mean age inferred from tracer data with the 'real' mean (ideal) age is important for the understanding and interpretation of tracer derived ages. A correct understanding of them helps to detect changes in ocean ventilation and, e.g. to infer anthropogenic carbon or ocean utilization rates from transient tracer data. This study provides a significant contribution to this topic, although the model analyses is restricted to the isopycnal sigma_theta=25.5.

The text is clear and well written, whereas the figures could need some improvement.

General comments:

For the case of constant Delta/Gamma ratios, the values 0f 0.8, 1.0, 1.2 and 1.4 are chosen. When inferring Delta/Gamma from CFC-12 and SF6, the color bar reaches from 0.2 to 1.8 (the same range has been used in He et al. 2018 to infer anthropogenic carbon from IG functions). Why is the range of the assumed Delta/Gamma ratios so much smaller (one could choose e.g. 0.2, 0.6, 1.0. 1.4 and 1.8)? (For the case Delta/Gamma=1.8 I would expect that the IG derived mean age is larger than the ideal age at least for the earlier years.)

A: We thank you for your thoughtful and constructive feedback on our work. We limited the Delta/Gamma for single-tracer constrained IG-TTD to 0.8 and 1.4 since the spatially constant ratio is considered as mean of a distribution varying in space and in time including the extremes (e.g., 0.2). Therefore, the ratio around 1 turns to be a good averaged approximation (He et al., 2018) and has been widely used in many other studies (e.g., Waugh et al., 2004, 2006; Tanhua et al., 2008; Jeansson et al., 2020, 2023). Instead of applying the extreme value of Delta/Gamma =0.2 globally, for the dual-tracer constrained IG-TTD we accepted that for some regions, the Delta/Gamma can be very low (e.g., 0.2).

In this study, only absolute values for the differences and temporal changes in age are presented. This implies, that difference between ideal and tracer derived age values and a temporal change in the ages is weighted equally, independent from the age value itself. I wonder whether this is appropriate. For young waters (Gamma~5 yr), an age change of +- 2 years over the considered time period or a difference between ideal and tracer derived age of ~2 yr is significant, whereas for old waters (Gamma~100 yr), such changes/differences would be negligible. I would thus suggest to also calculate relative age differences between ideal and tracer derived ages and also relative changes of age over time. If the results for the relative age changes/differences do not substantially differ from the absolute changes/differences presented here, this should be mentioned in the text. Otherwise, the relative age changes/differences need be discussed in addition to the absolute changes.

A: We thank the reviewer for this suggestion and have provided an additional figure showing the percent mismatch between mean age with different assumptions in the delta/gamma and ideal age (by using (mean age)/(ideal age) x 100-100, Fig S2). To be brief, besides the old waters in the northeast tropical Pacific and the Bay of Bengal with Delta/Gamma values higher than 1.0, the mean age of IG-TTD underestimates the ideal age. Moreover, the relative difference between ideal age and mean age is slightly higher in young waters.

[Figure]

Fig. S2: Panel (a) shows the distribution of ideal age (yr) averaged from 1981 to 2015 on isopycnal layer $\sigma_0$= 25.5 kg/m3 in the esm-piControl simulation. Panels (b-f) show the percent mismatch between mean age with different assumptions about the value of delta/gamma and the simulated ideal age (by using (mean age)/(ideal age) $\times$ 100-100).

The analyses focuses on the globally integrated/averaged mean age of the tracer inferred IG functions, i.e. the global distribution is 'condensed' to one number. This implies that regional differences might cancel out (e.g. the trend in age and the difference between tracer derived mean age and ideal age could differ between regions and even have opposite signs). In reality, also age changes for specific regions (e. g. upwelling, or water mass formation regions) are of interest, not only globally averaged values. Therefor, I would suggest to show at least one map with the differences between tracer derived and ideal age (for the 'best' tracer derived mean age), and also one map with the differences in the temporal trend between tracer derived ('best' result) and ideal age.

A: We thank you for your suggestion and totally understand your concern. We have added a map showing the temporal trend of ideal age and tracer-derived age (Fig 9) and discussed it in the revised manuscript:

"Noteworthy, the dual-tracer constrained IG-TTD demonstrates superior performance in discerning spatial patterns and magnitude of temporal changes in ideal age compared to the single-tracer constrained IG-TTD (Fig.9). The single-tracer constrained IG-TTD is very sensitive to the chosen value of Delta/Gamma, commonly showing a spurious age increase with low values of Delta/Gamma and an age decrease with high values of Delta/Gamma across all ocean basins (Fig.9c-f). While the dual-tracer method exhibits some spurious trends in the eastern tropical Atlantic and western tropical Indian Ocean, it generally provides a more accurate representation of the spatial patterns and magnitudes of true ideal age trends. Notably, it correctly identifies regions with no significant trends in ideal age (Fig.9a,b)."

[Figure]

Fig.9: Temporal trend of ideal age (a) and mean age of IG-TTD with different assumptions on the value of delta/gamma (b-f) on isopycnal layer $\sigma_0$= 25.5 kg/m3 in the *esm-hist* simulation. Stippling designates areas where the ratio of standard deviation and mean of the regression slope of ideal age (or mean age) against time exceeds 1 (i.e., no significant trends).

Why is the mean age not inferred by calculating the Delta/Gamma ratio from SF6 and CFC-12 at every grid point for every year?

The results presented here are based on spatially variable Delta/Gamma ratios, but without

temporal change (the Delta/Gamma ratios from the years 2000, 2005, 2010 and 2015 are applied to the whole time series). Maybe the age calculation with the actual, time varying Delta/Gamma ratios could even replace the four differnt age calculations presented here.

A: We appreciate the reviewer's idea of using a time-varying constrained Delta/Gamma ratio. However, we here also focus on how to reconstruct the past ventilation change as long as possible based on the available measurements. If we constrain the Delta/Gamma ratio according to where and also when we have observations of both CFC-12 and SF6, we would have to limit our temporal analysis mostly to the period after 2000 rather than starting in 1981 since between 1981 and 2000 only CFC-12 measurements are available. We have clarified this in the revised manuscript. Our results suggest that it is reasonable to apply the Delta/Gamma ratio constrained in specific years to all previous or afterward measurements (at least within a few decades investigated here). We also re-evaluated and used this technique in a follow-on manuscript on "Variation of ventilation in the North Atlantic over the past three decades - a climate change signal" (https://www.researchsquare.com/article/rs-5595029/v1 )

Specific comments:

l. 119-120 and Fig. 1

Why has the isopycnal 26.0+-0.5 been chosen? The whole analyses is restricted to the isopycnal 25.5,

wouldn't it be more reasonable to show the data for this isopycnal (25.5 +-0.5) here?

A: We thank the reviewer for pointing this out, and we have modified the figure showing the data for isopycnal 25.5 +- 0.5 kg/m3 and modified the paragraphs describing it.

[Figure]

Fig. 1: Distribution of (a) subsampled simulated CFC-12, (b) observed CFC-12 mixing ratio, and (c) their difference on the isopycnal layer $\sigma_0 = 25.5 \pm 0.5$ kg $\cdot$ m$^{-3}$ , with the unit of parts per trillion (ppt).

l.323-327

The results from the study in Peacock and Maltrud (2006) are interpretated wrongly.

First, the IG derived CFC values are smaller (half of) than the CFC values derived from the 'real'

TTD (actual value). This would imply that the IG derived TTD is too old compared to the real TTD, in l. 326-327 the opposite is stated.

Second, the mean age of the 'real' and of the IG TTD are identical, because the parameters Delta and Gamma for the IG function are derived by calculating mean age and width of the 'real' TTD. Hence, it is wrong to say the mean age of the IG TTD differs from the 'real' water age. The difference is that the IG function contains a smaller fraction of young water, thus the inferred CFC values are smaller and the water 'appears' older compared to the 'real' TTD. One could also conclude that the shape of the 'real' TTD in this case differs significantly from the shape of an IG function.

A: We agree that TTD can have different shapes (e.g., contain multimodal), and the IG shape might lose some information. We thank the reviewer for this comment and would like to explain why we still think Peacock and Maltrud (2006) implicitly suggested that IG-TTD tends to underestimate the real mean age of TTD (or ideal age) with modified sentences below.

For example, Peacock and Maltrud (2006)  compared the concentration of CFC-like tracer derived by convolution of ocean surface CFC boundary condition with the model-simulated TTD ("actual" CFC) and the one derived by the same boundary condition with the IG-TTD ("predicted" CFC). Both TTD and IG-TTD share the same Gamma and Delta. They found that "predicted" CFC-like concentrations are only half of "actual" values at a depth of 245 meters in the tropical regions (see their Fig.~13). In other words, to achieve the same abiotic transient tracer concentration inferred from model-simulated TTD, the IG-TTD mean age would need to be reduced,  i.e., the mean age of IG-TTD is smaller than the mean age of "real" TTD with the same partial pressure of CFC.

l.326

'directly simulated CFC-like tracer'

This is misleading, as the CFC-like tracer in Peacock and Maltrud (2006) is inferred from the modeled TTD (convolution integral of TTD and assumed tracer surface concentration). This is not what I understand as 'directly simulated'.

Regarding the difference between tracer derived and 'real' TTDs also the study from Chouksey et al. (2022) could be mentioned. There, tracer inferred IG-TTDs are compared with TTDs inferred from modeled (numerical) floats for the AAIW range. In some cases, the tracer derived TTDs are younger, in some cases older than the float based TTDs. Also, the shape of the float derived TTDs sometimes differs from the shape of an IG function.

A: We really appreciate this correction and are sorry that we misunderstood this part from Peacock and Maltrud (2006). We have corrected this in the revised manuscript. We clarified this: "Peacock and Maltrud (2006)  compared the concentration of CFC-like tracer derived by convolution of ocean surface CFC boundary condition with the model-simulated TTD ("actual" CFC) and the one derived by the same boundary condition with the IG-TTD  ("predicted" CFC). Both TTD and IG-TTD share the same Gamma and Delta."

We also enjoyed the reading of Chouksey et al. (2022), who compared the simulated float-derived TTD shape and the one applying the IG function. We thank the reviewer for this very appropriate reference and have added it to our revised manuscript.

l.327-328

Here, the study from Steinfeldt et al. (2024) could be cited. These authors found an increase of age (and hence a negative anomaly of anthropogenic carbon) with time for the old deep waters of the Atlantic when parameterizing the TTD as a single IG function. Assuming a contribution of an additional 'old' TTD leads to smaller age changes over time.

A: We thank the reviewer for this relevant reference and have added it to our revised manuscript.

l.351-352

'...and also from the cut-off of the long-tail of old ages in the spectrum due to the limited

atmospheric history length of CFC-12 and SF6'

This is not true, as the IG-functions always include a long tail towards high ages. This tail is more pronounced for higher Delta/Gamma ratios, leading to the increase of the mean age with the Delta/Gamma ratio. What is true is that this tail cannot be constrained from CFC-12 and SF6, as is correctly stated in l. 335. Please rephrase.

A: We thank the reviewer for pointing out this, and we have modified the sentence as "Such a difference might arise from the assumption that the transit-time distribution of water parcels follows the unimodal Inverse Gaussian distribution and from the limited atmospheric history length of CFC-12 and $SF_6$ which cannot constrain well the long tail towards high ages in the spectrum. "

Figures:

In general, the maps showing global distributions are too small. This could be changed easily, e.g.:

Fig.1: placing the color bar below the figures and showing latitude labels only on the left figure

would allow to increase the maps itself significantly

A: We have modified Fig. 1 as suggested.

Fig. 2 and 5:

These figures stretch over one whole page, but there is a lot of free space between the single maps, which could be enlarged

A: We have modified our figures according to suggestions as shown below. For Fig. 2, we combined panels (d) and (e) into one panel and removed the ample free space. For Fig. 5, we

combined mean and specific years constrained Delta/Gamma in *esm-piControl* simulation and *esm-Hist* simulation.

[Figure]

Fig.2: Distribution of (a) ideal age (yr), (b) apparent oxygen utilization (AOU, mmol·m⁻³), and (c) depth (meter) averaged from 1981 to 2015 at isopycnal layer $\sigma_0$=25.5 kg·m⁻³ in *esm-piControl* simulations. Waters shallower than the local winter mixing depth have been excluded. Panel (d) presents the temporal evolution of ideal age (red line, with left y-axis) and AOU (blue line, with right y-axis) averaged at the isopycnal layer $\sigma_0$=25.5 kg·m⁻³ in *esm-piControl* simulation.

[Figure]

Fig. 5: Delta/Gamma constrained by the simulated concentration of CFC-12 and $SF_6$ on isopycnal layer $\sigma_0$=25.5 kg·m$^{-3}$ under pre-industrial and historical forcing conditions. Panels (c,e,g,i) show Delta/Gamma constrained in 2000, 2005, 2010, and 2015, minus the temporal mean Delta/Gamma (panel a) in the *esm-piControl* experiment. Panels (d,f,h,j) show Delta/Gamma$ constrained in 2000, 2005, 2010, and 2015, minus the temporal mean Delta/Gamma (panel b) in the *esm-Hist* experiment. During the calculation, we assume 100% surface saturation of both tracers.

Fig.4:

latitude labels could be omitted at the right figure

The labels at the color bar are 'cut off' at the right side (the same holds for figure 2b)

A: We have modified our figures according to suggestions.

Fig. 3a, 6a and 7a: quantity and unit for the color bar are missing

the correlation values 0.95 and 0.9? at the Taylor Diagram overlap with the color bar

A: We have added the unit for the color bar and overcome the overlap between the Taylor Diagram and the color bar. We show Fig. 3 as an example here.

[Figure]

Fig. 3: In the pre-industrial control run, panel (a) presents the Taylor Diagram between 1981 to 2015 averaged mean age of IG-TTD and the ideal age (as reference) at isopycnal layer $\sigma_0$=25.5 kg·m$^{-3}$ and the color pattern provides the bias in %. The symbols of star, diamond, triangle and circle indicate Delta/Gamma as 0.8, 1.0, 1.2 and 1.4 respectively are applied in IG-TTD calculation. Panel (b) shows the global-averaged ideal age (black), and mean age with Delta/Gamma of 0.8 (red), 1.0 (green), 1.2 (cyan), and 1.4 (blue) from 1981 to 2015.

Minor comments:

Title: 'water mass' in two words

A: We have modified the word according to suggestions.

Additional literature:

Chouksey, M., Griesel, A., Eden, C. and Steinfeldt, R. (2022), Transit Time Distributions and ventilation pathways using CFCs and Lagrangian backtracking in the South Atlantic of an eddying ocean model. J. Phys. Oceanogr., 52(7), 1531-1548, 2022, doi:10.1175/JPO-D-21-0070.1.

Steinfeldt, R., Rhein, M., and Kieke, D. (2024), Anthropogenic carbon storage and its decadal changes in the Atlantic between 1990–2020, Biogeosciences, 21, 3839–3867, doi:10.5194/bg-21-3839-2024.